# CSDE1 controls gene expression through the miRNA-mediated decay machinery

Pavan Kumar Kakumani[1,2] , Louis-Mathieu Harvey[1,2], François Houle[1,2], Tanit Guitart[3], Fátima Gebauer[3,4], Martin J Simard[1,2]

In animals, miRNAs are the most prevalent small non-coding RNA molecules controlling posttranscriptional gene regulation. The Argonaute proteins (AGO) mediate miRNA-guided gene silencing by recruiting multiple factors involved in translational repression, deadenylation, and decapping. Here, we report that CSDE1, an RNA-binding protein linked to stem cell maintenance and metastasis in cancer, interacts with AGO2 within miRNA-induced silencing complex and mediates gene silencing through its N-terminal domains. We show that CSDE1 interacts with LSM14A, a constituent of P-body assembly and further associates to the DCP1–DCP2 decapping complex, suggesting that CSDE1 could promote the decay of miRNA-induced silencing complex-targeted mRNAs. Together, our findings uncover a hitherto unknown mechanism used by CSDE1 in the control of gene expression mediated by the miRNA pathway.

## Introduction

miRNAs play important roles in many physiological processes dictating the timing of development and pathogenesis of multiple diseases in animals (1). They are a major class of small non-coding RNAs of 21–23 nt in length that repress their target mRNAs in a sequence-specific manner through their association with the Argonaute (AGO) proteins (2, 3). The miRNA duplex is initially loaded onto AGO and the passenger strand is released thus guiding the miRNA:AGO complex to complementary sites located primarily in 3′ UTRs. Once the miRNA–mRNA interaction is instigated, AGO recruits GW182 (known as TNRC6 in humans) which in turn interacts with poly(A)-binding protein (PABP) and the deadenylase complex, namely, CCR4–NOT comprising CNOT1 and CCR4 among others, to shorten the poly(A) tail of the mRNA (4, 5, 6, 7). The deadenylated 3′-terminus serves as a binding platform for numerous protein factors promoting translational repression of the mRNA and facilitating the removal of the 5′-terminal cap structure by the DCP1–DCP2 decapping complex

(8). Subsequently, the decapped mRNAs are degraded by the 5′-3′ exonuclease XRN1 and removed from the translational pool (9). However, a recent study demonstrated the uncoupling of translational repression from target mRNA decay dependent on tissue specificity in animal models (10). Recruitment of the decapping complex to the 5′-terminal cap structure is orchestrated by an intricate, dynamic network of protein–protein interactions involving various decapping factors and translational repressors. These proteins are generally localized to discrete cytoplasmic loci called processing (P) bodies (11), which include enhancer of decapping 3 and 4 (EDC3 and EDC4), the eIF4E-binding protein 4E-T, Like Sm14 (LSM14), and the DEAD-box RNA helicase DDX6 (12). Although deadenylation most often precedes mRNA decapping, examples do exist of mRNAs that undergo degradation independent of deadenylation (13).

Cold shock domain (CSD)–containing proteins belong to the most evolutionarily conserved family of RNA-binding proteins (RBPs). So far, a select number of these RBPs have been shown to participate in miRNA-mediated gene silencing. For example, LIN28 and DIS3L2, target let-7 miRNA precursors and interfere with the biogenesis of let-7 to maintain pluripotency of mESCs (14, 15). In fact, the CSD present in the N terminus of LIN28A and DIS3L2 plays a critical role in either binding specific members of the let-7 family or degrading the uridylated version of them, respectively (16, 17).

CSDE1, also known as upstream of N-Ras (UNR), is one member of this family containing at least five CSDs (18). CSDE1/UNR is known as a regulator of translation and mRNA stability in various organisms (18, 19). In *Drosophila melanogaster*, this protein is part of a translational repressor complex assembled at the 3′ UTR of *male-e-specific-lethal-2* (*msl-2*) mRNA and is critical for the proper regulation of X-chromosome dosage compensation (20, 21). In mammalian cells, CSDE1 binds to its own internal ribosome entry site to repress translation (22) but also enhances internal ribosome entry site–dependent translation of select transcripts, such as Apaf1 (apoptotic peptidase activating factor 1) and Cdk11B (cyclin-dependent kinase 11B) in co-operation with neuronal polypyrimidine tract–binding protein and hnRNPC1/C2, respectively (23, 24, 25). CSDE1 was also shown to interact with PABP to control the

[1]St-Patrick Research Group in Basic Oncology, Centre Hospitalier Universitaire de Québec-Université Laval Research Center, L'Hôtel-Dieu de Québec, Québec City, Canada   [2]Laval University Cancer Research Centre, Québec City, Canada   [3]Gene Regulation, Stem Cells and Cancer Programme, Centre for Genomic Regulation, Barcelona Institute of Science and Technology, Barcelona, Spain   [4]Universitat Pompeu Fabra, Barcelona, Spain

Correspondence: martin.simard@crchudequebec.ulaval.ca; pavan-kumar.kakumani@crchudequebec.ulaval.ca

stability of c-fos mRNA, and with AUF1 to regulate that of PTH mRNA (26). In melanoma cells, it binds Vimentin (VIM) and RAC1 mRNAs to increase their translation and trigger metastasis (27). Besides, CSDE1 has the capacity to modulate the same biological process in an opposing manner depending on the cell type and state (28, 29, 30). For instance, CSDE1 promotes erythroblast differentiation, whereas it maintains an undifferentiated state both in human embryonic stem cells and mouse embryonic stem cells by preventing their commitment to neuroectoderm or primitive endoderm, respectively (29, 30). In the former case, CSDE1 regulates the stability and translation of FABP7 and VIM mRNAs, whereas in the latter, it controls Gata6 posttranscriptionally. Overall, it is intriguing to notice that CSDE1 displays a versatile behavior depending on the context. However, the molecular understanding of its mechanisms promoting such opposite regulatory roles is still incomplete.

In the present study, we used small RNA affinity purification and identified CSDE1 as a new component of the miRNA-induced silencing complex (miRISC) that contributes to miRNA-mediated gene silencing. Analysis of protein–protein interactions show that the N-terminal domains of CSDE1 facilitate the interaction of AGO2 with components of P-bodies as well as interact with the mRNA decay machinery. Taken together, our data provide compelling evidence that CSDE1 is a novel effector of miRNA-mediated gene silencing.

## Results

### Interaction of CSDE1/UNR with miRISC is conserved among animals

To uncover proteins interacting with miRNAs in mESCs, we purified proteins associated with miR-20a-5p (expressed at optimal levels in mESCs) from cell lysate using modified complementary probes. Upon mass spectrometry analysis, we identified CSDE1 to be associated with the miRISC. To test if CSDE1 interaction with miRISC was specific to mESCs and to miR-20a-5p, we performed pull-downs using lysates prepared from mESCs, fibroblasts, and human embryonic kidney cell line including another miRNA, miR-295-3p. Similar to miR-20a-5p, miR-295-3p pull-down enriched CSDE1 as well as AGO2 (Fig 1A), and as these interactions are constantly seen in different cell types, it is suggested that CSDE1 is a new bona fide component of miRISC. Because CSDE1 is conserved among animals and its homolog (UNR) plays an important role in *Drosophila* development (20), we sought to examine the involvement of UNR with the miRNA pathway in flies. We used *Drosophila* embryo extract (DEE) and performed pull-down assays for select miRNAs that control development of *D. melanogaster* (31). As shown in Fig 1B, the protein UNR interacts with all the tested miRISC. Furthermore, as small RNA pathways in *Drosophila* use different AGO proteins to deliver their respective outcome on gene silencing (32), we examined the interaction between UNR and the miRNA specific AGO1 in DEE and observed that UNR is associated with AGO1 in insects (Fig 1B). Taken together, these results reveal that UNR/CSDE1 is a new component of miRISC conserved among animals.

### CSDE1 interacts with AGO2 through its N-terminal domains

As previous reports showed that some RBPs can interact with the miRISC in an RNA-dependent manner through their binding to mRNAs (33), we thus wanted to determine if CSDE1 interacts directly or through RNA molecules with the miRISC. We pulled-down the miR-20a-5p miRISC as well as immunoprecipitated endogenous AGO2 from cell

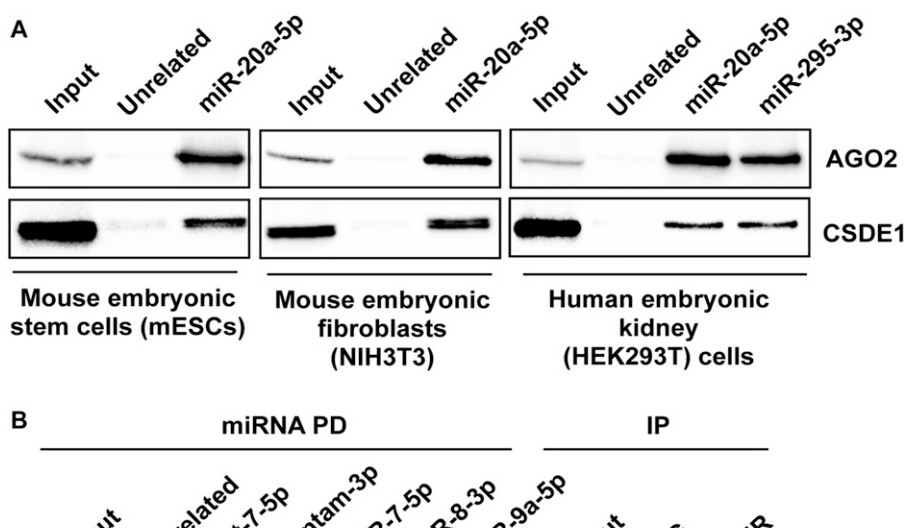

**Figure 1. CSDE1 interacts with different miRISC in animals.**
**(A)** Western blot analysis of miRNA pull-downs using 2′-O-methyl–biotinylated oligonucleotides complementary to listed miRNAs from the lysate of mESCs (V6.5), NIH3T3, and HEK293T cells. Representative immunoblots of four independent experiments are shown. **(B)** Left panel: 2′-O-methyl pull-down of multiple miRNAs from *Drosophila* embryos extracts followed by Western blot detection of UNR and *Drosophila* AGO1 (dAGO1). Right panel: co-immunoprecipitation of UNR and dAGO1. The data are representative of three independent experiments.

lysates treated with RNases and monitored the presence of CSDE1 in purified complexes. We observed that in both the cases, the association of CSDE1 with both purified complexes is retained even after the depletion of RNA molecules (Fig 2A), indicating that the interaction of CSDE1 with the miRISC is not bridged by mRNAs.

Mammalian CSDE1 proteins consist of five CSDs. The C-terminal domain after the CSDs is essential to bind the UNR-interactor protein (UNRIP) but otherwise dispensable to bind RNA and 4E-T (34, 35, 36). To identify which domain of CSDE1 is responsible for interaction with AGO2, we generated deletion mutants as shown in the schematics of Fig 2B. The recombinant FLAG-tagged clones were transiently expressed in cells and immunoprecipitation was performed using FLAG antibody. As shown in Fig 2C, the N-terminal deletion mutant ΔCSD1 lost its interaction with AGO2, suggesting CSD1 is required for its association to AGO2.

### The CSD 2 of CSDE1 is necessary for optimal silencing of miRNA targets

Because CSDE1 interacts with miRISC through its essential component AGO2, we probed whether CSDE1 is required to mediate silencing of miRNA targets. To investigate the potential role for CSDE1 in miRNA-mediated gene silencing, we used an miRNA activity reporter in which the 3′UTR of *Renilla reniformis* luciferase (RL) harbors eight bulged binding sites for let-7, mimicking the mode of a typical miRNA binding. This reporter system provides a sensitive measurement for the miRNA-mediated gene silencing activity. When the reporter system was tested in cells depleted of CSDE1 expression, the luciferase levels were increased when compared with the control, indicating that the loss of CSDE1 resulted in impairment of let-7 silencing activity (Fig 3A). When the let-7 target sites were removed from the luciferase reporter, this differential effect on silencing was abolished between the Control and CSDE1-depleted cells (Fig 3A), confirming the specificity of the observed effects of CSDE1 on miRNA activity. Next, to determine whether the domains responsible for CSDE1 interaction with AGO2 are necessary for the gene silencing, we transiently expressed the N-terminal FLAG-CSDE1 and its deletion mutants (ΔCSD1 and ΔCSD12) in CSDE1-depleted background and performed the miRNA reporter assays. As shown in Fig 3B, there was a significant difference in the luciferase activity in presence of

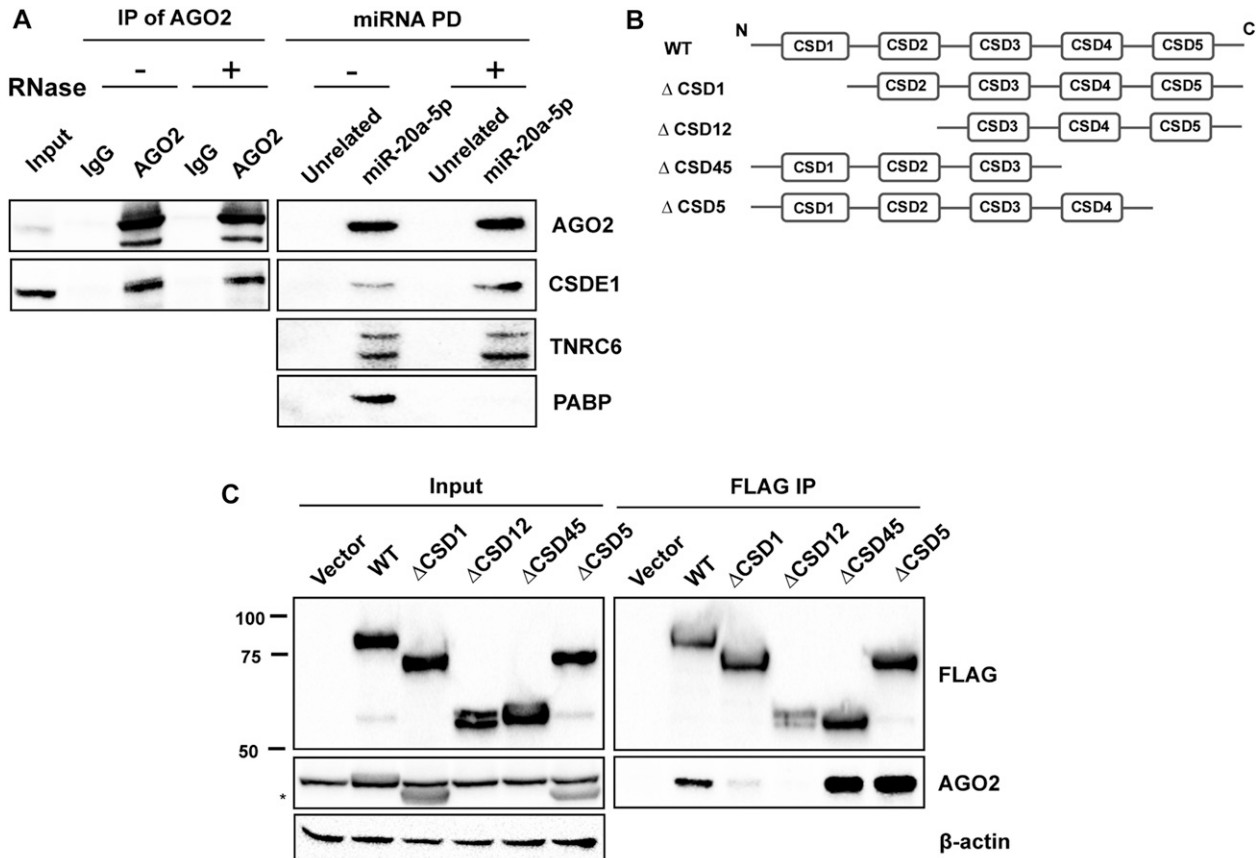

**Figure 2.   CSDE1 interacts with AGO2 through its N-terminal domains.**
**(A)** Immunoprecipitation of endogenous AGO2 using anti-AGO2 antibody (left) and miR-20a-5p miRNA pull-down (PD) from the lysate of NIH3T3 cells, before (−) and after (+) RNases treatment. The samples were run on the SDS–PAGE gel and probed with the antibodies indicated. Same cell extracts were used to perform both immunoprecipitation (IP) of AGO2 and miR-20a-5p pull-down (PD). Representative data of two independent experiments are shown. **(B)** Schematics of the deletion mutants of CSDE1 used in the study. **(C)** Immunoprecipitation of CSDE1 using anti-FLAG antibody from the lysate of HEK293T cells transiently expressing FLAG-tagged wild-type (WT) and deletion mutants of CSDE1. The immunoprecipitates were run on the SDS–PAGE gel and the proteins indicated were probed. β-actin was used as a loading control. The data are representative of four independent experiments. Migration of the molecular weight marker is indicated (kD). The star (*) denotes nonspecific band.

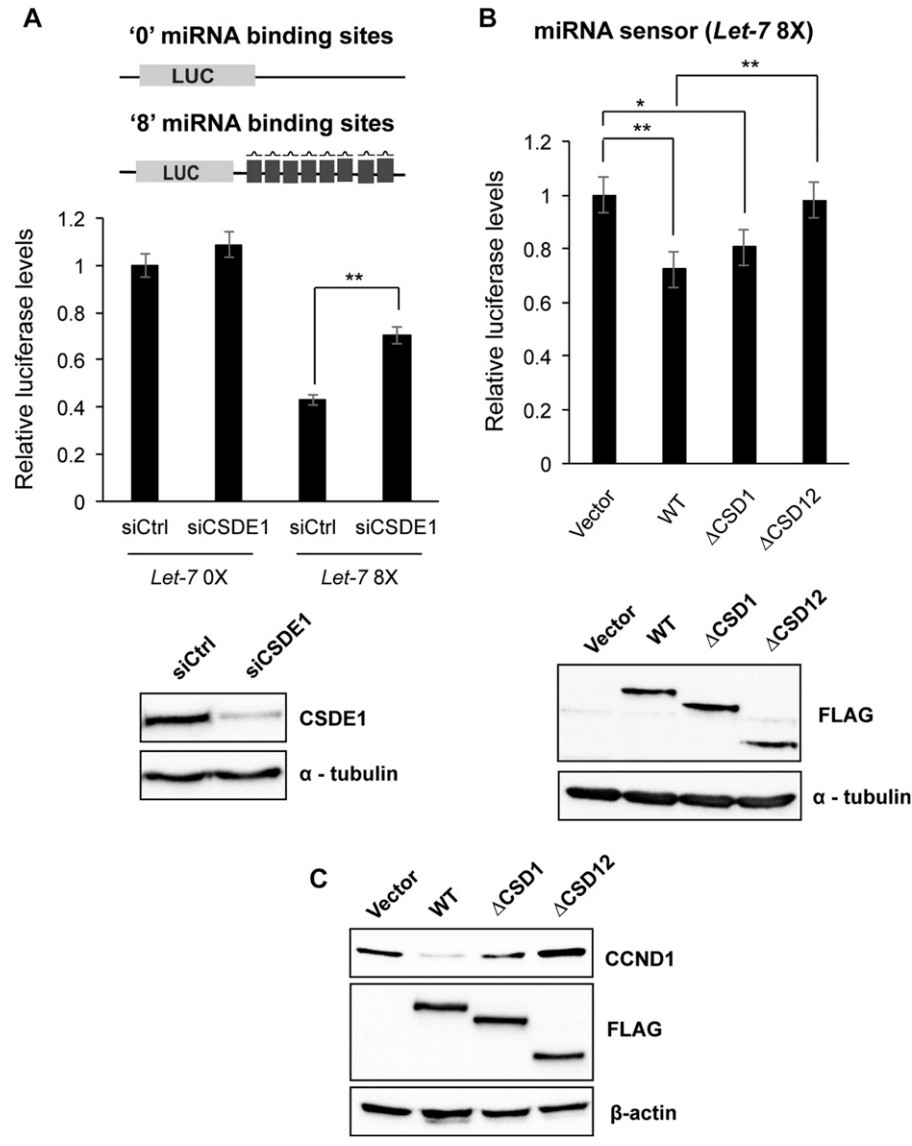

**Figure 3. CSDE1 influences luciferase activity of miRNA reporter constructs.**
**(A)** Upper panel: relative luciferase levels (*Renilla*/firefly) of miRNA reporter constructs with "0" (left) and "8" (right) let-7 binding sites detected upon Control and CSDE1 knockdown conditions in NIH3T3 cells. Lower panel: Western blot confirming the expression of CSDE1 in Control (siCtrl) versus CSDE1 knockdown (siCSDE1) conditions in NIH3T3 cells. **(B)** Upper panel: relative luciferase levels (*Renilla*/firefly) of miRNA reporter construct with "8" binding sites in shCSDE1 NIH3T3 cells transiently expressing the deletion mutants of CSDE1 as indicated. Lower panel: representative figure for the Western blot confirming the expression of FLAG-tagged CSDE1 and its mutants in shCSDE1 NIH3T3 cells used for the luciferase assays. **(C)** Western blot confirming the expression of CCND1 in shCSDE1 HEK293T cells transiently expressing the deletion mutants of CSDE1 as indicated. β-actin was used as a loading control. Immunoblots representative of two independent experiments are shown. In (A, B), data are presented as mean ± SEM. (**P* < 0.03, ***P* < 0.01 from three independent experiments; two tailed *t* test).

the deletion mutant ΔCSD12 when compared with either the Wt or ΔCSD1 proteins. The expression of CSDE1 and its deletion mutants for the reporter assays were comparable, as presented in Fig 3B. Later, we validated our findings from reporter assays by checking the expression of one of miR-20a-5p endogenous targets, namely, cyclin D1 (CCND1) (37) (Fig 3C). Furthermore, to ascertain that the effects of CSDE1 in the control of gene expression are mediated through miRISC, we artificially tethered CSDE1 to the 3′UTR of luciferase transcripts carrying or not a terminal poly (A) tail and measured reporter activity. In contrast to tristetraprolin (TTP) and the suppressor domain of TNRC6, two proteins that silence gene expression when tethered to mRNA (5, 38), there was no significant difference in luciferase activity of both reporters for tethered CSDE1 when compared with negative control, suggesting that CSDE1 interaction with the AGO2/miRISC is important to mediate gene silencing (Fig S1A–C). Taken together,

these results indicate that CSDE1, in particular, its N-terminal domain CSD2, is required for miRNA-mediated gene silencing in mammalian cells.

### CSDE1 tethers AGO2 to the P-body component LSM14A

To decipher the role of CSDE1 in the function of miRISC, we first examined whether AGO2 binding to miRNA was affected by CSDE1. Here, we immunoprecipitated endogenous AGO2 from control as well as CSDE1-depleted P19 stem cells and measured the levels of miR-20a-5p present in the pellet. Our results showed no significant difference between the samples (Fig S2A), indicating CSDE1 does not interfere with steady-state levels of miRNAs bound to AGO2. Next, we tested the interaction between CSDE1 and TNRC6, a scaffolding protein critical for the assembly of miRISC (4, 5, 6), as well as evaluated the contribution of CSDE1

for the binding of proteins associated with miRNA-loaded AGO2. We observed that CSDE1 does not interact with TNRC6, and there was no difference in the association of the proteins TNRC6, CNOT1, and PABP to either miR-20a-5p or AGO2 in CSDE1-depleted cells compared with control (Fig S2B and C), suggesting that CSDE1 has no role in the formation of the core miRISC. We then examined the interactome of both endogenous AGO2 and CSDE1 proteins. As shown in Fig 4A, both AGO2 and CSDE1 interact with LSM14A, a crucial protein for P-body assembly (38, 39). Next, we examined whether CSDE1 mediates the interaction between AGO2 and LSM14A. As shown in Fig 4B, the interaction between AGO2 and LSM14A is considerably reduced in CSDE1 knockdown conditions when compared with the control. Later, to determine which of the CSD domains is responsible for CSDE1 interaction with LSM14A, the N-terminal FLAG-CSDE1 and its deletion mutants (ΔCSD1, ΔCSD12, ΔCSD45, and ΔCSD5) were transiently expressed in cells, and immunoprecipitation was performed using FLAG antibody. We observed that the ΔCSD12 deletion mutant, but not ΔCSD1 alone, lost the interaction with LSM14A, indicating CSD2 is accountable for interaction between CSDE1 and LSM14A (Fig 4C). Furthermore, we even tested whether CSDE1 interaction with AGO2 is dependent on LSM14A. As shown in Fig S3, the alteration of LSM14A expression does not influence the binding of AGO2 to CSDE1. Together, our results confirm that CSDE1 facilitates the interaction of AGO2-miRISC with the components of P-body for subsequent target repression.

## CSDE1 interacts with the mRNA decapping complex

CSDE1 was earlier shown to interact with CCR4 to control the levels of select target mRNAs (40). Because CCR4 is recruited to miRISC through the CCR4–NOT complex, we investigated whether CSDE1 is involved in CCR4 interactions with the miRISC. Here, we performed immunoprecipitations against transiently expressing AGO2 and endogenous CNOT1 in CSDE1 repressed background. As observed in Fig S4A and B, CSDE1 has no effect on the binding of CCR4 to either AGO2 or CNOT1, essential components of miRISC and CCR4–NOT complex, respectively. Next, we investigated whether CSDE1 plays a role in recruiting the DCP1–DCP2 decapping complex to CCR4–NOT complex. We first examined the association between CSDE1 and the components of DCP1–DCP2 decapping complex and observed that EDC3, EDC4, DCP1α, and DCP2 interact with CSDE1 in an RNA-independent manner similar to its ubiquitous interacting partner, UNRIP (Figs S4C and 5A; a difference in interaction intensity between CSDE1 and EDC4 is observed in different cell lines). Subsequently, we surveyed the domain responsible for CSDE1 interaction with EDC3 and EDC4 which are responsible for the association of decapping complex to CNOT1 and recruiting the mRNA-decapping enzyme DCP2 (41, 42). We found no association of either EDC3 or EDC4 with the ΔCSD12 deletion mutant (Fig 5B), indicating that the CSD2 domain is essential for CSDE1 interaction with the decapping machinery. We also examined the association between CCR4–NOT complex and components of the decapping complex in the absence of CSDE1 and observed that CSDE1 has no role in

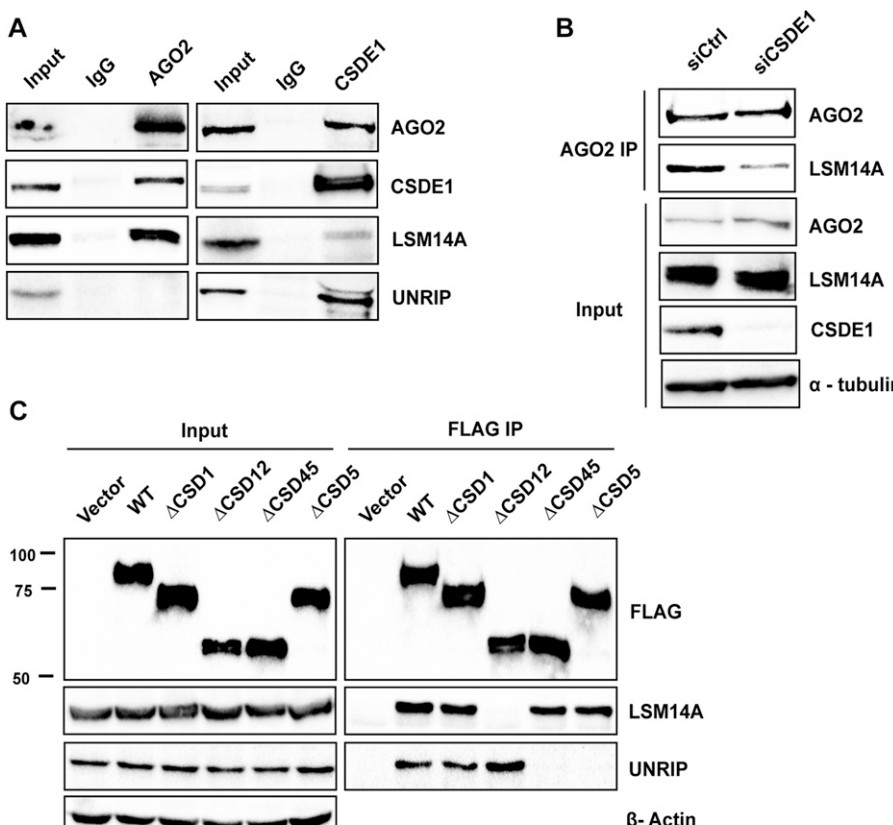

**Figure 4.  CSDE1 mediates interaction between AGO2 and LSM14A.**
**(A)** Immunoprecipitation of AGO2 and CSDE1 using respective antibodies from NIH3T3 cells while IgG was used as a control. The immunoprecipitates were run on the SDS–PAGE gel and probed for the indicated proteins. **(B)** Immunoprecipitation (IP) of AGO2 using anti-AGO2 antibody was performed from the lysate of P19 stem cells under Control (siCtrl) and CSDE1 knockdown (siCSDE1) conditions. The samples were run on the SDS–PAGE gel and probed with the antibodies indicated. α-tubulin was used as a loading control. The IP and input data were generated from the same cell extracts. **(A, B)** The data are representative of two independent experiments. **(C)** Immunoprecipitation (IP) of CSDE1 was performed using anti-FLAG antibody from the lysate of HEK293T cells transiently expressing FLAG-tagged wild-type (WT) and deletion mutants of CSDE1. The immunoprecipitates were run on the SDS–PAGE gel and the proteins indicated were probed. Migration of the molecular weight marker is indicated (kD). β-actin was used as a loading control. The data are representative of four independent experiments.

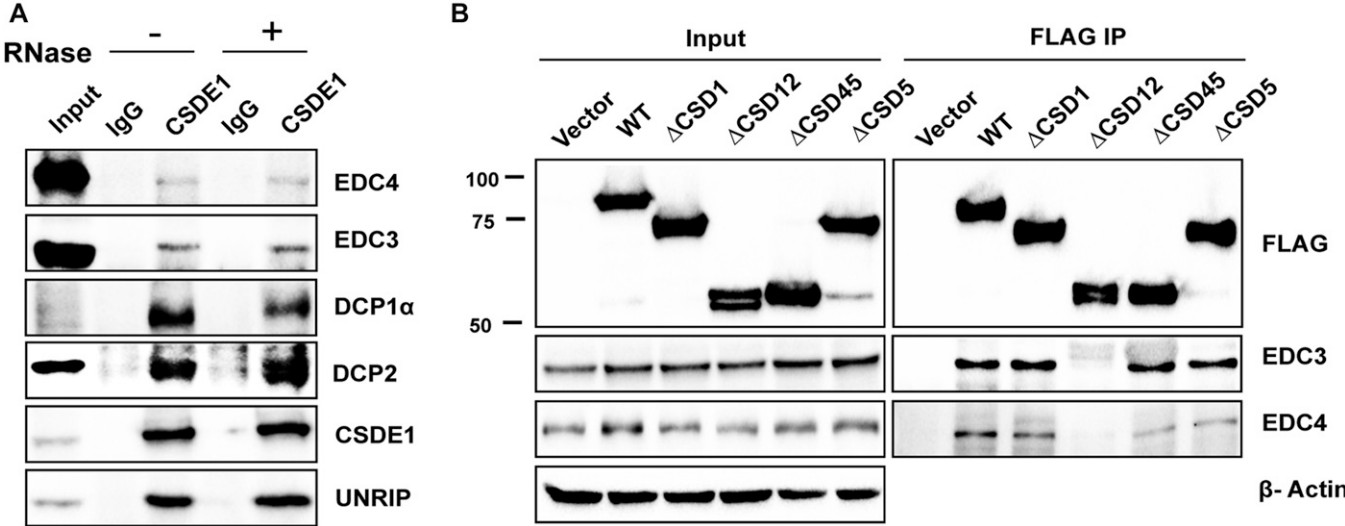

**Figure 5. CSDE1 associates with mRNA decapping machinery.**
**(A)** Immunoprecipitation of endogenous CSDE1 using anti-CSDE1 antibody from the lysate of HEK293T cells, before (−) and after (+) RNases treatment. The samples were run on the SDS–PAGE gel and probed with the antibodies indicated. The data representative of two independent experiments are shown. **(B)** Immunoprecipitation of CSDE1 was performed using anti-FLAG antibody from the lysate prepared from shCSDE1 HEK293T cells transiently expressing FLAG-tagged wild-type (WT) and deletion mutants of CSDE1. The immunoprecipitates were run on the SDS–PAGE gels and the proteins indicated were probed. β-actin was used as a loading control. The data are representative of four independent experiments.

facilitating the interaction between the decapping proteins and CNOT1 (Fig S4B). Collectively, our results suggest that CSDE1 is an interacting partner of decapping complex.

## Discussion

RBPs and miRNAs target mRNAs on 3′UTR to facilitate posttranscriptional gene regulation. miRNAs mediate gene silencing through miRISC, composed of multiple RBPs, including AGO2. The dynamics of interaction between miRISC components define the outcome of translational repression and/or decay of targeted mRNAs. CSDE1 is an RBP that has been shown to bind specific mRNAs at various regions and, depending on target and cellular context, it either promotes or hinders translation, accompanied by destabilization of the transcript. Here, we established an interaction between CSDE1 and miRISC, especially AGO2, implying a role for CSDE1 in the control of gene expression mediated by miRNAs. The N-terminal domain CSD1 is crucial for the binding of CSDE1 to AGO2. Because CSD1 is also important for RNA binding of CSDE1 (38), we hypothesize that the interaction between CSDE1 and AGO2 may involve mRNA before the loading of CSDE1 onto miRISC, and that once loaded, this interaction becomes RNA independent. A similar scenario has been observed for the interaction of UNR and maleless (MLE) in *Drosophila*. Here, the interaction requires roX2 RNA but, once established, it becomes resistant to treatment with RNases (43).

CSDE1 promotes miRNA-mediated silencing of reporter mRNAs and an endogenous miRNA target CCND1. As the CSDE1 ΔCSD1 mutant, which lost interaction with AGO2, behaves similarly to the WT CSDE1 in miRNA sensor assay, it is likely that the strong repression caused by the presence of multiple miRNA-binding sites

found on the reporter masks the effects of this CSDE1 mutant. Furthermore, CSDE1 does not alleviate gene expression when tethered to the mRNA 3′UTR which suggests that CSDE1 effects on gene silencing are dependent on miRNA binding to the target mRNA and its cumulative association with interacting partners besides AGO2. In sharp contrast, *Drosophila* UNR has been shown to repress translation when tethered to the 3′ UTR of *msl-2* transcripts (34), and CSDE1 has been recently shown to promote the translation of select oncogenic mRNAs through binding to their 3′UTRs in melanoma cells (27), suggesting that the regulatory capacity of CSDE1 depends on the specific binding context and/or cellular factors. Similar to CSDE1, sequence-specific RBPs, namely, HuR, Pumilio (PUM1 and PUM2) and FAM120A were shown to exert a promoting and/or antagonistic effect on AGO2-mediated miRNA-guided control of gene expression (44, 45, 46). In light of these reports, it would be interesting to study whether RBPs with physiological and pathological roles such as CSDE1 might be selecting miRNA targets and thus influencing the activity of miRISC on respective mRNAs.

The AGO2:miRNA binding to mRNAs promote their localization to P-bodies through their interaction with TNRC6 proteins (4, 5, 6, 7). Our results show that CSDE1 neither interacts with TNRC6 nor influences its association with AGO2/miRISC. We also observed that CSDE1 has no role in the repression of targets artificially tethered to AGO2 (Fig S1D and E). Because the repression of target mRNAs artificially tethered to AGO2 is dependent of TNRC6 (47), it is not surprising that the depletion of CSDE1 does not affect this assay. Besides, CSDE1 mediates the interaction between AGO2 and LSM14A, which is critical for P-body assembly (39). But, loss of LSM14A did not abrogate the interaction between CSDE1 and AGO2 suggesting that P-body assembly is not a prerequisite of this interaction or for miRNA-mediated gene silencing. This is in agreement with earlier

reports that P-body formation is rather an effect than a cause for miRNA-mediated gene regulation (48). Interestingly, it adds up to our notion that CSDE1 interaction with LSM14A makes it a tether for holding the miRISC onto P-bodies. This opens up the possibility that CSDE1 interaction with P-body components provides the opportunity to form molecular condensates where the necessary factors that mediate deadenylation and decapping are concentrated for more efficient repression. Besides, it was earlier reported that CSDE1 interacts with LSM14A through its C terminus via UNRIP while associated to 4E-T at the N terminus (35). In contrast, our results indicate a UNRIP-independent association of CSDE1 to LSM14A, which suggest that a new surface of interactions appear when CSDE1 interacts with AGO2 and their involvement in miRNA-mediated target repression are important questions for future research.

In miRNA-mediated decay of target mRNAs, the miRISC recruits CCR4–NOT complex to the 3'UTR of targeted mRNA, which acts as a platform for the binding of the DCP1:DCP2 decapping complex, promoting mRNA decay. As observed from our results, CSDE1 interacts with miRISC and CCR4. However, those interactions have no influence on recruiting the deadenylating enzyme either to AGO2 or CNOT1. Rather, CSDE1 interacts with the components of the DCP1–DCP2 decapping complex. Because CSDE1 interacts with 4E-T (35), we speculate that CSDE1 might bridge the 3'-terminal mRNA decay machinery with the mRNA 5' terminal cap structure via its interactions with 4E-T and AGO2. This would, in turn, increase the local concentration of DCP2 around the cap structure, promoting mRNA decay which needs further validation.

Overall, the mechanistic details elucidated in the present study not only highlight the importance for CSDE1 in the control of miRNA-mediated expression but also opens up a new dimension for studies on cold shock proteins in translational control and mRNA storage. Our findings provide an alternate mechanism used by cold shock proteins in the miRNA pathway, rising the functional scope of these proteins beyond the biogenesis level as demonstrated in the case of LIN28 (15, 16, 17). Moreover, our data might explain the behavior of CSDE1 in maintaining cell stemness by destabilizing mRNAs. Understanding the mechanisms used by CSDE1 may impinge on future studies on other cold shock proteins, such as the Y-Box family of proteins (YBX1, YBX2, and YBX3) and CSD-like proteins with translational (EIF1AX, EIF2A, and EIF5A) and mRNA processing (DHX8 and EXOSC3) activities for their plausible involvement in miRNA-mediated gene regulation and stem cell homeostasis.

# Materials and Methods

### Cell culture and transfection

mESCs V6.5 were grown on gelatin (0.2%)-coated plates in the 2iLIF medium (Neurobasal 240 ml, DMEM/F12 244 ml, 0.25× N2-Supplement, 0.5× B27 Supplement minus Vitamin A, BSA Fraction V 0.167 ml, 2 mM L-Glutamine, 50 U/ml penicillin, and 50 µg/ml streptomycin). P19 mouse stem cells (American Type Culture Collection, CRL-1825) were grown according to the

supplier's instructions. NIH3T3, HeLa, and HEK293T cells were grown in DMEM supplemented with 10% fetal bovine serum, 50 U/ml penicillin, and 50 µg/ml streptomycin.

### Plasmids

The N-terminal FLAG-CSDE1 and its deletion mutants were generated by cloning respective DNA fragments into BamHI and XhoI sites in pCDNA-FLAG vector. The shRNA plasmids used to generate conditional knockdown (cKD) cell lines for both control and CSDE1 were described previously (27). psiCHECK2-*Let-7* 8X and psiCHECK2-*Let-7* 0X were described previously (49). In psiCHECK2 (Promega), *Renilla* luciferase is used as the primary reporter gene and the miRNA-binding sites are cloned into the multiple cloning region located downstream of the *Renilla* translational stop codon. The vector also contains a second reporter gene, firefly luciferase which allows normalization of *Renilla* luciferase expression. Full-length CSDE1 was cloned into EcoRI and NotI sites in pCI-λNHA plasmid for tethering experiments. pCI-λNHA-LacZ, pCI-λNHA-AGO2, and reporter plasmids (RL-5BoxB, RL-5BoxB-A114-N40- HhR, and FL) were described previously (47, 50, 51).

### Antibodies

Antibodies against CSDE1, AGO2, DCP1α, PABP, LSM14A, and DCP2 are from Abcam. The CNOT1 antibody was from Proteintech. The TNRC6, CCR4, and UNRIP antibodies are from Novus Biologicals. FLAG, HA antibodies are from Sigma-Aldrich and Cell Signaling, respectively. The UNR antibody was generated in-house at Dr Gebauer's laboratory. The dAGO1 antibody was received from Dr Yuki Tomari's laboratory, Tokyo University, Japan. The AGO2 antibody (11A9) was received from Dr Gunter Meister laboratory, University of Regensberg, Germany.

### 2'-O-methyl (2'-O-Me) pull-down

The cells were homogenized in the lysis buffer (25 mM Tris–HCl, pH 7.4, 150 mM NaCl, 1% NP-40, 1 mM EDTA, and 5% glycerol) with protease and phosphatase inhibitor cocktail (Thermo Fisher Scientific). The lysate was centrifuged at 15,000$g$ for 15 min, the supernatant was collected and measured for protein concentration using Bradford reagent (Bio-Rad). Next, 8 mg of cell lysate was precleared with M-280 Streptavidin Dynabeads (Thermo Fisher Scientific) and nonspecific biotinylated 2'-O-Me oligonucleotides (called Unrelated, 10 pmol) for 40 min at RT with rotation. The supernatant was incubated with respective biotinylated 2'-O-Me oligonucleotides (10 pmol) bound to streptavidin beads for 40 min at RT. The beads were washed three times using ice-cold lysis buffer. For RNase treatment, the RNases A&T1 were added in buffered solution to the beads and incubated at RT for 15 min. Of note, the miRNAs bound by the AGO proteins are resistant to the treatment (52). The beads were resuspended in 1× SDS loading buffer and eluted by heating at 100°C for 2 min before loading on to the SDS–PAGE gel and analyzed by Western blotting. The complementary oligonucleotide probes used for the pull-down assays are as follows:

Unrelated: 5′-Bio-CAUCACGUACGCGGAAUACUUCGAAAUGUC-3′
has-miR-20a-5p: 5′-Bio-UCUUCCUACCUGCACUAUAAGCACUUUAACCUU-3′
hsa-miR-295-3p: 5′-Bio-AGACUCAAAAGUAGUAGCACUUU-3′
dme-let-7-5p: 5′-Bio-UCUUCACUAUACAACCUACUACC-3′
dme-bantam-3p: 5′-Bio-UCUUC AAUCAGCUUUCAAAAUGAUCUCAACCUU-3′
dme-miR-7-5p: 5′-Bio-UCUUC ACAACAAAAUCACUAGUCUUCCAACCUU-3′
dme-miR-8-3p: 5′-Bio-UCUUC GACAUCUUUACCUGACAGUAUUAACCUU-3′
dme-miR-9a-5p: 5′-Bio-UCUUCUCAUACAGCUAGAUAACCAAAGAACCUU-3′.

### Co-immunoprecipitation

Cells were homogenized in the lysis buffer with protease and phosphatase inhibitor cocktail. The lysate was centrifuged at 15,000$g$ for 15 min, and the supernatant was collected and measured for protein concentration using Bradford reagent. Meanwhile, Dynabeads Protein G (Thermo Fisher Scientific) were prepared in an Eppendorf (25 $\mu$l for a total protein extract of 2 mg and 10 $\mu$g of antibody). The beads were washed with three times the volume of lysis buffer and the step was repeated two more times. The lysate was preincubated with the equivalent of Dynabeads at 4°C on a nutator for about 45 min. The lysate was collected, and the respective antibody added to incubate on nutator at 4°C for overnight. Next day, the Dynabeads (as earlier) were prepared and the lysate was added to incubate on the rotator at RT for about 90 min. The beads were washed about five times with lysis buffer and the samples were extracted by adding SDS loading buffer in 1× concentration and heating at 100°C.

In case of fly, Oregon-R *D. melanogaster* extracts were prepared from 0 to 16 h (overnight) embryos as described earlier (53). Co-immunoprecipitations were performed using 3 $\mu$g of purified anti-UNR or unspecific rabbit IgGs bound to Dynabeads protein A (Thermo Fisher Scientific). 13 $\mu$l of the corresponding bead slurry was incubated with 500 $\mu$g of DEE with complete protease inhibitor cocktail (Roche) and phosphate buffer. After 2 h of incubation at RT, six washes were performed with 10× bead volumes of cold 1× NET buffer (50 mM Tris–HCl, pH 7.5, 150 mM NaCl, 0.1% NP40, and 1 mM EDTA). Proteins were recovered with 1× SDS loading buffer and resolved on a SDS–PAGE gel.

### Luciferase reporter assays

For miRNA assays, NIH3T3 cells were grown to 60–80% confluency on 24-well plates. Per well, 500 ng of FLAG-CSDE1 and its mutant constructs were transfected using JetPRIME reagent (Polyplus transfection) according to the manufacturer's instructions. 24 h posttransfection, the cells were again transfected with 500 ng of the miRNA reporter constructs using the reagent. The cells were lysed 48 h posttransfection with 100 $\mu$l of 1× passive lysis buffer (Promega) and the reporter activity was measured using Luminoskan Ascent (Thermo Fisher Scientific). For tethering assays, HeLa and HEK293 cells were grown to 60–80% confluency on 24-well plates. Per well, 500 ng of λNHA-constructs were transfected using JetPRIME reagent (Polyplus transfection) according to the manufacturer's instructions. 48 h posttransfection, the cells were again transfected with 50 ng (100 ng for experiments made with HEK293 cells) each of *Renilla* and Firefly luciferase reporter constructs using the reagent. The cells were lysed 24 h posttransfection with 100 $\mu$l of 1× passive lysis

buffer (Promega), and the reporter activity was measured using Luminoskan Ascent.

### Generation of cKD cell lines

The cKD cell lines of NIH3T3 and HEK293T were generated as described previously (27). In brief, lentiviral constructs expressing shControl (shCtrl) or shCSDE1 were obtained from Dharmacon (TRIPZ shControl [RHS4743], TRIPZ shCSDE1 clone ID V2THS_212077 [RHS4696-200681476]). The cells were infected with lentiviral particles expressing shControl or shCSDE1. After selection with 1 $\mu$g/ml puromycin, co-expression of shRNA was induced by adding 1 $\mu$g/ml doxycycline to the medium.

# Supplementary Information

# Acknowledgements

We are grateful to Drs Marc Fabian, Gunter Meister, and Yukihide Tomari for providing various reagents used in this study and Dr Steve Bilodeau for initial help in setting up mESC culture. We also thank all members of our laboratory for helpful discussions and the mass spectrometry facility from the Centre Hospitalier Universitaire de Québec-Université Laval Research Center. Funding: The Canadian Institutes of Health Research supported this work (MJ Simard). PK Kakumani and LM Harvey are recipients of scholarship from the Fonds de Recherche du Québec-Santé (FRQ-S). MJ Simard is a Research Chair from FRQ-S. F Gebauer was supported by grants from the Spanish Ministry of Economy and Competitiveness (BFU2015-68741), the Catalan Government (2017 SGR 799), Marató de TV3 (TV'13-20131430), and the Centre of Excellence Severo Ochoa.

### Author Contributions

PK Kakumani: conceptualization, data curation, formal analysis, supervision, validation, investigation, visualization, methodology, and writing—original draft, review, and editing.
L-M Harvey: data curation and formal analysis.
F Houle: data curation and methodology.
T Guitart: conceptualization, resources, data curation, formal analysis, validation, investigation, visualization, and methodology.
F Gebauer: conceptualization, resources, formal analysis, supervision, funding acquisition, visualization, project administration, and writing—original draft, review, and editing.
MJ Simard: conceptualization, formal analysis, supervision, funding acquisition, project administration, and writing—original draft, review, and editing.

### Conflict of Interest Statement

The authors declare that they have no conflict of interest.

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
