## [Reviewer comments · Life Science Alliance]

Life Science Alliance

CSDE1 controls gene expression through the miRNA-mediated decay machinery

Pavan Kakumani, Louis-Mathieu Harvey, François Houle, Tanit Guitart, Fatima Gebauer, and Martin Simard

DOI: <https://doi.org/10.26508/lsa.201900632>

Corresponding author(s): Martin Simard, CHU de Quebec-Universite Laval Research Center and Pavan Kakumani, Oncology-CHU de Québec-Université Laval Research Centre (L'Hôtel-Dieu de Québec)

Review Timeline:

Submission Date:	2019-12-19
Editorial Decision:	2020-01-29
Revision Received:	2020-02-28
Editorial Decision:	2020-03-02
Revision Received:	2020-03-03
Accepted:	2020-03-04

Scientific Editor: Andrea Leibfried

Transaction Report:

January 29, 2020

Re: Life Science Alliance manuscript #LSA-2019-00632-T

Prof. Martin J Simard
CHU de Quebec-Universite Laval Research Center
9 McMahon
Quebec City G1R2J6
Canada

Dear Dr. Simard,

Thank you for submitting your manuscript entitled "CSDE1 controls gene expression through the miRNA-mediated decay machinery" to Life Science Alliance. The manuscript was assessed by expert reviewers, whose comments are appended to this letter. A third report was promised on your work, but since we did not receive it up to now, I am moving the process forward. Should the third reviewer provide input within the next week, I will forward the report to you. The below decision outcome will not get affected by the potential additional input.

As you will see, the two reviewers appreciate your data and provide constructive input on how to further strengthen your manuscript. We would thus like to invite you to submit a revised version to us, addressing the individual concerns raised. This seems rather straightforward, but please do get in touch in case you would like to discuss individual revision points further.

Thank you for this interesting contribution to Life Science Alliance. We are looking forward to receiving your revised manuscript.

Sincerely,

B. MANUSCRIPT ORGANIZATION AND FORMATTING:

Reviewer #1 (Comments to the Authors (Required)):

Here Kakumani and colleagues submit a manuscript identifying CSDE1 (cold shock domain containing E1) protein (also known as UNR) as a new interactor with AGO2 and the miRISC. Moreover, they provide evidence that CSDE1 interacts with the P-body protein LSM14A and the

mRNA decapping machinery, and mediates gene silencing via its N-terminal domains. CSDE1 was identified as an AGO2 associated protein by affinity pulldowns of miRISCs in various systems, including *Drosophila* embryo, mouse embryonic stem cell, fibroblast and HEK293 cell lysates. Using deletion mutants they determine that CSDE1 uses its N-terminal domains to interact with AGO2 in an RNA-independent manner. They go on to show that depleting CSDE1 via RNAi impairs miRNA-mediated gene silencing of a reporter RNA containing a number of let-7 miRNA target sites and that only CSDE1 proteins that interact with AGO2 mediate this effect. They further show that CSDE1 interacts with LSM14 and contributes to the localization of AGO2 to P-bodies. Finally, using co-IPs they demonstrate that CSDE1 interacts with the mRNA decapping complex, including DCP1, EDC4 and weakly with EDC3. Overall this is an interesting paper that identifies and characterizes a new AGO2-interacting protein in the context of miRNA-mediated gene silencing. It is well written, the data are of high quality and it will be well-received by the field.

Comments:

- 1) The authors mention on page 7 that they artificially tether CSDE1 to the 3'UTR of a luciferase transcript "carrying or not a poly(A) tail". This data was shown in Figure EV1. However, when I looked at their schematic diagrams of the constructs, both contain poly(A) tails. It's just that one construct contains an internalized poly(A) tail rather than one that is generated in vivo via polyadenylation. This needs to be clarified.
- 2) Is the effect CSDE1 has on miRNA silencing impacting the ability of the miRISC to interact with a target mRNA? This could be tested by seeing if depleting CSDE1 impacts the ability of artificially tethered AGO2 to silence a target mRNA in a miRNA-independent manner.
- 3) In Figure 2, the authors show that FLAG-tagged WT CSDE1 interacts with AGO2 (co-IP experiments). It would be informative to see if CSDE1 also co-precipitates TNRC6 proteins by western blotting. Also, I noticed that Figure 2C, there is no Actin blot for the FLAG-IP samples. This would be helpful to conclude that the interaction between CSDE1 and AGO2 is specific and not because CSDE1 is sticky.
- 4) The authors show in Figure 5 that CSDE1 co-precipitates EDC4 weakly, yet in Figure EV5C this interaction is much stronger. Can the authors comment on this inconsistency?
- 5) The authors provide immunofluorescence images in EV3 and EV4A to support a conclusion that CSDE1 is important for AGO2 to localize with LSM14 in P bodies. However, this data is not super convincing. While AGO2 co-localizes with LSM14 in EV3, it does not seem to co-localize with DCP1 in EV4A but is rather diffuse when CSDE1 is overexpressed. Does CSDE1 overexpression lead to AGO2 still properly colocalizing in punctate foci with LSM14? Overall, while this data is not the strongest, I'm not sure that it is necessary for the paper. The functional conclusions and interaction data in other figures is still important, whereas the P-body localization data doesn't add a huge amount in my opinion.

Reviewer #2 (Comments to the Authors (Required)):

The manuscript entitled "CSDE1 controls gene expression through the miRNA-mediated decay machinery" characterized CSDE1/UNR as novel cofactor of animal miRISC. Through experiments based at culture cells, the authors provided sufficient evidence to support their conclusion. CSDE1 is known to regulate gene expression posttranscriptionally by interacting with diverse mRNAs. The regulatory target and effect of CSDE1 are specific depending on cellular contexts and

developmental stages. The authors' findings connecting CSDE1 and miRNA silencing complexes might be relevant to the CSDE1's functional versatility in vivo. Together with Lin28 and Dis3L2, the unraveled property of CSDE1 highlights that CSD-containing proteins are involved in miRNA biogenesis and actions.

Major point.

It was suggested that CSD1 domain is required for CSDE1-AGO2 interaction (Fig. 2), while the delta CSD1 mutant was as functional as WT CSDE1 in the repression of miRNA sensor mRNA (Fig 3. Panel B). The authors should discuss about the inconsistency.

Minor points

Page 3.

"CSD containing" to be "Cold Shock Domain (CSD)-containing".

"family of RBPs" to be "family of RNA-binding proteins (RBPs)".

"let-7" to be *italic*. Other "let-7" also.

Page 4.

"Vimentin and RAC1" to be *italic*.

"FABP7 and VIM" to be *italic*.

Page 5.

Please specify the strand of miR-20a (5P or 3P), as authors have done to 295-3p.

Same to be mentioned to other miRNAs targeted in pull-down experiments in Fig. 1.

"Similar to AGO2, we...(Fig 1A)" does not explain the result.

For example, "Similar to miR-20a, miR-295-3p enriched CSDE1 as well as AGO2, and these interactions are constantly seen in different cell types".

Page 6.

"the interaction of CSDE1 with the miRISC is direct.."

Even if CSDE1-miRISC interaction is resistant to RNase treatment, it does not necessarily mean the binding is "direct". Perhaps the binding is bridged by other proteins.

Page 7.

"mature miRNA mediated" to be "miRNA-mediated".

"loading of miRNAs" to be "loading and stability of miRNAs" or "steady-state miRNA levels".

Page 8.

"deltaCSD12 deletion mutant lost.." to be "deltaCSD12 deletion mutant, but not deltaCSD1 deletion alone, lost.."

"with CSDE1 in a RNA independent" to be "with CSDE1 in an RNA-independent".

Page 9.

"partner of miRISC-recruited decapping complex". It was not addressed if the decapping factors are recruited by miRISC or not in Fig. 5. "miRISC-recruited" to be deleted. Alternatively, the authors might test the interaction between CSDE1 and decapping complex in AGO2 deficiency.

"RNA binding proteins" to be "RNA-binding proteins".

"a direct interaction between". "direct" to be removed.

Page 10.

"sequence specific RNA binding" to be "sequence-specific RNA-binding".

"UNRIP independent" to be "UNRIP-independent".

Page 11.

"miRNA mediated" to be "miRNA-mediated".

We thank both reviewers for their insightful inputs on our manuscript. Their comments clearly improved our manuscript. Here are the detailed answers to specific questions of the reviewers:

Reviewer #1

Here Kakumani and colleagues submit a manuscript identifying CSDE1 (cold shock domain containing E1) protein (also known as UNR) as a new interactor with AGO2 and the miRISC. Moreover, they provide evidence that CSDE1 interacts with the P-body protein LSM14A and the mRNA decapping machinery, and mediates gene silencing via its N-terminal domains. CSDE1 was identified as an AGO2 associated protein by affinity pulldowns of miRISCs in various systems, including *Drosophila* embryo, mouse embryonic stem cell, fibroblast and HEK293 cell lysates. Using deletion mutants they determine that CSDE1 uses its N-terminal domains to interact with AGO2 in an RNA-independent manner. They go on to show that depleting CSDE1 via RNAi impairs miRNA-mediated gene silencing of a reporter RNA containing a number of let-7 miRNA target sites and that only CSDE1 proteins that interact with AGO2 mediate this effect. They further show that CSDE1 interacts with LSM14 and contributes to the localization of AGO2 to P-bodies. Finally, using co-IPs they demonstrate that CSDE1 interacts with the mRNA decapping complex, including DCP1, EDC4 and weakly with EDC3. Overall this is an interesting paper that identifies and characterizes a new AGO2-interacting protein in the context of miRNA-mediated gene silencing. It is well written, the data are of high quality and it will be well-received by the field.

We are thanking the Reviewer for her/his appreciation of the quality and the relevance of this study for our field.

Comments:

1) The authors mention on page 7 that they artificially tether CSDE1 to the 3'UTR of a luciferase transcript "carrying or not a poly(A) tail". This data was shown in Figure EV1. However, when I looked at their schematic diagrams of the constructs, both contain poly(A) tails. It's just that one construct contains an internalized poly(A) tail rather than one that is generated in vivo via polyadenylation. This needs to be clarified.

We thank the Reviewer for pointing that out: the schematic diagram is right. We have now modified the description in the text by:

"Further, to ascertain that the effects of CSDE1 in the control of gene expression are mediated through miRISC, we artificially tethered CSDE1 to the 3'UTR of luciferase transcripts carrying or not a terminal poly(A) tail and measured reporter activity."

2) Is the effect CSDE1 has on miRNA silencing impacting the ability of the miRISC to interact with a target mRNA? This could be tested by seeing if depleting CSDE1 impacts

the ability of artificially tethered AGO2 to silence a target mRNA in a miRNA-independent manner.

We have performed the experiment proposed by the Reviewer and included it in the revised version (new Figure S1D-E). Our new data show that the depletion of CSDE1 does not impact the silencing of artificially tethered AGO2. As TNRC6 does not interact with CSDE1 (new Figure S2), we explain those new observations in the Discussion section as follows:

“The AGO2:miRNA binding to mRNAs promote their localization to P-bodies through their interaction with TNRC6 proteins [4-7]. Our results show that CSDE1 neither interacts with TNRC6 nor influences its association with AGO2/miRISC. We also observed that CSDE1 has no role in the repression of targets artificially tethered to AGO2 (Fig S1D-E). Since, the repression of target mRNAs artificially tethered to AGO2 is dependent of TNRC6 [47], it is not surprising that the depletion of CSDE1 does not affect this assay.”

3) In Figure 2, the authors show that FLAG-tagged WT CSDE1 interacts with AGO2 (co-IP experiments). It would be informative to see if CSDE1 also co-precipitates TNRC6 proteins by western blotting. Also, I noticed that Figure 2C, there is no Actin blot for the FLAG-IP samples. This would be helpful to conclude that the interaction between CSDE1 and AGO2 is specific and not because CSDE1 is sticky.

We have now added the data showing that TNRC6 does not co-precipitate with CSDE1 (new Figure S2B), suggesting that CSDE1 is not a “sticky” protein.

4) The authors show in Figure 5 that CSDE1 co-precipitates EDC4 weakly, yet in Figure EV5C this interaction is much stronger. Can the authors comment on this inconsistency?

This is likely because we are using different cell lines in those experiments. We have now clarified this in the text by adding the following on Page 8:

*“We first examined the association between CSDE1 and the components of DCP1-DCP2 decapping complex and observed that EDC3, EDC4, DCP1 α and DCP2 interact with CSDE1 in an RNA-independent manner similar to its ubiquitous interacting partner, UNRIP (Fig S4C, 5A; **a difference in interaction intensity between CSDE1 and EDC4 is observed in different cell lines**).”*

5) The authors provide immunofluorescence images in EV3 and EV4A to support a conclusion that CSDE1 is important for AGO2 to localize with LSM14 in P bodies. However, this data is not super convincing. While AGO2 co-localizes with LSM14 in EV3, it does not seem to co-localize with DCP1 in EV4A but is rather diffuse when CSDE1 is overexpressed. Does CSDE1 overexpression lead to AGO2 still properly colocalizing in

punctate foci with LSM14? Overall, while this data is not the strongest, I'm not sure that it is necessary for the paper. The functional conclusions and interaction data in other figures is still important, whereas the P-body localization data doesn't add a huge amount in my opinion.

We agree with the reviewer that those data are not necessary. We therefore decided to remove them from the manuscript.

Reviewer #2:

The manuscript entitled "CSDE1 controls gene expression through the miRNA-mediated decay machinery" characterized CSDE1/UNR as novel cofactor of animal miRISC. Through experiments based at culture cells, the authors provided sufficient evidence to support their conclusion. CSDE1 is known to regulate gene expression posttranscriptionally by interacting with diverse mRNAs. The regulatory target and effect of CSDE1 are specific depending on cellular contexts and developmental stages. The authors' findings connecting CSDE1 and miRNA silencing complexes might be relevant to the CSDE1's functional versatility in vivo. Together with Lin28 and Dis3L2, the unraveled property of CSDE1 highlights that CSD-containing proteins are involved in miRNA biogenesis and actions.

We are thanking the Reviewer for her/his appreciation of the the relevance of this study.

Major point.

It was suggested that CSD1 domain is required for CSDE1-AGO2 interaction (Fig. 2), while the delta CSD1 mutant was as functional as WT CSDE1 in the repression of miRNA sensor mRNA (Fig 3. Panel B). The authors should discuss about the inconsistency.

This is a good point raised by the Reviewer. We have now addressed this in the Discussion section as follows:

“As CSDE1 Δ CSD1 mutant, which lost interaction with AGO2, behaves similarly to the WT CSDE1 in miRNA sensor assay, it is likely that the strong repression caused by the presence of multiple miRNA binding sites found on the reporter masks the effects of this CSDE1 mutant. Further, CSDE1 does not alleviate gene expression when tethered to the mRNA 3'UTR which suggests that CSDE1 effects on gene silencing are dependent on miRNA binding to the target mRNA and its cumulative association with interacting partners besides AGO2.”

Minor points
Page 3.

"CSD containing" to be "Cold Shock Domain (CSD)-containing".

"family of RBPs" to be "family of RNA-binding proteins (RBPs)".

"let-7" to be *Italic*. Other "let-7" also.

Page 4.

"Vimentin and RAC1" to be *Italic*.

"FABP7 and VIM" to be *Italic*.

Page 5.

Please specify the strand of miR-20a (5P or 3P), as authors have done to 295-3p. Same to be mentioned to other miRNAs targeted in pull-down experiments in Fig. 1.

"Similar to AGO2, we...(Fig 1A)" does not explain the result.

For example, "Similar to miR-20a, miR-295-3p enriched CSDE1 as well as AGO2, and these interactions are constantly seen in different cell types".

Page 6.

"the interaction of CSDE1 with the miRISC is direct.."

Even if CSDE1-miRISC interaction is resistant to RNase treatment, it does not necessarily mean the binding is "direct". Perhaps the binding is bridged by other proteins.

Page 7.

"mature miRNA mediated" to be "miRNA-mediated".

"loading of miRNAs" to be "loading and stability of miRNAs" or "steady-state miRNA levels".

Page 8.

"deltaCSD12 deletion mutant lost.." to be "deltaCSD12 deletion mutant, but not deltaCSD1 deletion alone, lost.."

"with CSDE1 in a RNA independent" to be "with CSDE1 in an RNA-independent".

Page 9.

"partner of miRISC-recruited decapping complex". It was not addressed if the decapping factors are recruited by miRISC or not in Fig. 5. "miRISC-recruited" to be deleted. Alternatively, the authors might test the interaction between CSDE1 and decapping complex in AGO2 deficiency.

"RNA binding proteins" to be "RNA-binding proteins".

"a direct interaction between". "direct" to be removed.

Page 10.

"sequence specific RNA binding" to be "sequence-specific RNA-binding".

"UNRIP independent" to be "UNRIP-independent".

Page 11.

"miRNA mediated" to be "miRNA-mediated".

Thank you for pointing those out. We have corrected the typos and made the modifications listed by the Reviewer.

March 2, 2020

RE: Life Science Alliance Manuscript #LSA-2019-00632-TR

Prof. Martin J Simard
CHU de Quebec-Universite Laval Research Center
9 McMahon
Quebec City G1R2J6
Canada

Dear Dr. Simard,

Thank you for submitting your revised manuscript entitled "CSDE1 controls gene expression through the miRNA-mediated decay machinery". I now assessed the revisions performed and appreciate the introduced changes. I would thus be happy to publish your paper in Life Science Alliance, pending final revisions necessary to meet our formatting guidelines:

- please indicate the number of replicates performed throughout the manuscript
- please make sure that all corresponding authors link their profile in our submission system with their ORCID iD

A. FINAL FILES:

-- Summary blurb (enter in submission system): A short text summarizing in a single sentence the study (max. 200 characters including spaces). This text is used in conjunction with the titles of papers, hence should be informative and complementary to the title. It should describe the context and significance of the findings for a general readership; it should be written in the present tense

and refer to the work in the third person. Author names should not be mentioned.

B. MANUSCRIPT ORGANIZATION AND FORMATTING:

Sincerely,

March 4, 2020

RE: Life Science Alliance Manuscript #LSA-2019-00632-TRR

Prof. Martin J Simard
CHU de Quebec-Universite Laval Research Center
9 McMahon
Quebec City G1R3S3
Canada

Dear Dr. Simard,

Thank you for submitting your Research Article entitled "CSDE1 controls gene expression through the miRNA-mediated decay machinery". It is a pleasure to let you know that your manuscript is now accepted for publication in Life Science Alliance. Congratulations on this interesting work.

DISTRIBUTION OF MATERIALS:

Again, congratulations on a very nice paper. I hope you found the review process to be constructive and are pleased with how the manuscript was handled editorially. We look forward to future exciting submissions from your lab.

Sincerely,
